# FROM AMATEUR TO MASTER: INFUSING KNOWLEDGE INTO LLMS VIA AUTOMATED CURRICULUM LEARNING

## ABSTRACT

Large Language Models (LLMs) excel at general tasks but underperform in specialized domains like economics and psychology, which require deep, principled understanding. To address this, we introduce ACER (Automated Curriculum-Enhanced Regimen) that transforms generalist models into domain experts without sacrificing their broad capabilities. ACER first synthesizes a comprehensive, textbook-style curriculum by generating a table of contents for a subject and then creating question-answer (QA) pairs guided by Bloom's taxonomy. This ensures systematic topic coverage and progressively increasing difficulty. The resulting synthetic corpus is used for continual pretraining with an interleaved curriculum schedule, aligning learning across both content and cognitive dimensions.

Experiments with Llama 3.2 (1B and 3B) show significant gains in specialized MMLU subsets. In challenging domains like microeconomics, where baselines struggle, ACER boosts accuracy by 5 percentage points. Across all target domains, we observe a consistent macro-average improvement of 3 percentage points. Notably, ACER not only prevents catastrophic forgetting but also facilitates positive cross-domain knowledge transfer, improving performance on non-target domains by 0.7 points. Beyond MMLU, ACER enhances performance on knowledge-intensive benchmarks like ARC and GPQA by over 2 absolute points, while maintaining stable performance on general reasoning tasks. Our results demonstrate that ACER offers a scalable and effective recipe for closing critical domain gaps in LLMs.

## 1 INTRODUCTION

Large Language Models (LLMs) have achieved remarkable success across a wide range of natural language processing (NLP) tasks, including open-domain question answering, summarization, reasoning, and code generation, largely driven by scaling both model size and training data (Kaplan et al., 2020). However, this broad competency masks a critical weakness. While state-of-the-art models excel in general tasks, they falter in specialized domains that demand deep, principled understanding (He et al., 2025; Zhang et al., 2025b). This gap is highlighted in knowledge benchmarks like MMLU (Hendrycks et al., 2021), where performance in niche sub-domains like virology degrades, compared to broader medicine, revealing a gap between general knowledge and deep expertise.

This performance gap stems from the nature of LLM pretraining corpora, which are dominated by general web text and underrepresent specialized knowledge (Najem-Meyer et al., 2025). Even when domain-specific data is present, it lacks the methodical exposition and progressive knowledge scaffolding of expert materials like textbooks or lecture notes. Consequently, even large-scale models struggle with the technical terminology and hierarchical concepts essential for expert-level performance (Mai et al., 2024). Several strategies have been explored to address this gap. Instruction tuning improves alignment but not knowledge depth (Wei et al., 2022), while domain-specific pretraining often suffers from data scarcity and catastrophic forgetting of general capabilities (Gururangan et al., 2020; Béthune et al., 2025). Synthetic data generation has recently emerged as a promising alternative, with approaches such as Self-Instruct (Wang et al., 2023) and GLAN (Li

et al., 2025) showing clear benefits. However, these methods often lack systematic coverage of domain concepts or structured alignment, limiting their effectiveness in building principled domain expertise. This leaves a critical need for an approach that can instill deep, methodical knowledge without undermining the broad capabilities that make LLMs so powerful.

To address this challenge, we introduce ACER (Automated Curriculum-Enhanced Regimen), a framework designed to infuse LLMs with domain expertise without sacrificing their broad applicability. Our methodology consists of two core components: a process for systematically generating expert "study materials" and a novel training regimen for the model to absorb them. It begins by generating a detailed table of contents (ToC) that serves as the blueprint for textbook construction. The content is then generated section by section, resulting in a comprehensive synthetic textbook. Guided by Bloom's taxonomy (Bloom et al., 1956), ACER then complements synthetic textbooks with exam-style QA pairs to ensure systematic coverage and progressive difficulty. To reflect educational progression, we generate four versions of each textbook tailored to different audiences: high school, undergraduate, graduate, and researcher. The resulting synthetic corpora combine topical breadth with structured progression, enabling LLMs to acquire principled domain-specific knowledge through systematic curriculum progression that parallels human educational development. This synthetic curriculum is then used to continually pretrain a foundational LLM. We employ an interleaved training schedule across cognitive and content dimensions (**Cog+Con**) that strategically mixes the new expert corpora with general-domain data, enabling the model to gain deep expertise while retaining its broad capabilities.

Our evaluation methodology begins by systematically identifying a model's most significant knowledge gaps. To this end, we benchmarked the Llama 3.2 1B and 3B (Grattafiori et al., 2024) "student" models against their Llama 3.1 8B "teacher" across all 56 MMLU domains. The five domains exhibiting the most severe performance degradation were then selected as the proving ground for ACER's ability to build targeted expertise. We generated synthetic book corpora for these domains and continually pretrained the baseline models on this corpus, mixing synthetic data with general-domain replay data. To ablate different curriculum effects, we experimented with multiple scheduling strategies, including cognitive ordering (textbook → easy QA → hard QA) and persona-based content ordering (high school → undergraduate → graduate → researcher).

**Strong Results:** ACER consistently outperformed baselines, with cognitive and content based curriculum yielding macro-average gains of about 3 percentage points in the target domains. Specifically in challenging areas such as microeconomics, ACER improved accuracy by nearly 5 points. Additionally, performance across non-target domains also improved by about 0.7 points, suggesting cross-domain transfer without regressing from baseline accuracies. Beyond MMLU, we further evaluated ACER on widely used benchmarks such as ARC (Clark et al., 2018), GPQA (Rein et al., 2023), AGIEval (Zhong et al., 2024), GSM8K (Cobbe et al., 2021), and HellaSwag (Zellers et al., 2019). The knowledge-infused ACER-trained models improved by more than 2 absolute points in ARC and GPQA, both of which emphasize knowledge recall and domain understanding, while maintaining stable performance on general reasoning, arithmetic, and common sense tasks such as AGIEval, GSM8K, and HellaSwag. These results demonstrate that continual pretraining with ACER not only enhances specialized knowledge, but also preserves broad capabilities, providing a scalable recipe for closing domain gaps in LLMs.

Our contributions are:

1. **Systematic Synthesis of Expert Knowledge:** We introduce ACER, a framework that synthesizes structured textbook-style corpora and complementary exam-style QA pairs across multiple educational levels, enabling principled domain knowledge infusion into LLMs. (refer section 3)

2. **Effective Curriculum Learning Regimen:** We design and evaluate curriculum learning strategies (section 3.1) that align both cognitive and content dimensions. On MMLU, ACER delivers consistent macro-average improvements of about 3 percentage points across target domains, with particularly strong gains of up to 5 points in microeconomics. (refer section 4)

3. **Robust Generalization without Catastrophic Forgetting:** We demonstrate that ACER generalizes beyond in-domain tasks, yielding over 2 absolute point improvements on knowledge-intensive benchmarks such as ARC and GPQA. Crucially, these gains are

achieved without performance degradation on general reasoning, arithmetic, and common-sense tasks including AGIEval, GSM8K, and HellaSwag. (refer section 4.3)

## 2 RELATED WORK

LLMs have been very useful in general tasks, but they lag considerably in niche domains that demand deep, principled understanding (He et al., 2025; Zhang et al., 2025b). This performance gap is consistently reflected in benchmarks such as MMLU (Hendrycks et al., 2021), which span a wide range of specialized domains. One contributing factor is that specialized domains remain under-represented in the pretraining corpora (Najem-Meyer et al., 2025). Domain-adaptive pretraining has been effective in improving LLMs in such settings. Don't Stop Pretraining (Gururangan et al., 2020) demonstrated that continual pretraining on domain-specific corpora can significantly improve downstream task performance. Kerner (2024) further showed that even compact, specialized models can achieve competitive accuracy in-domain compared to much larger general-purpose LLMs. One reason, as suggested by (Mai et al., 2024), is that large general-purpose models struggle with domain-specific reasoning and hierarchical concepts essential for expert performance. More recent efforts, such as PreparedLLM (Chen et al., 2024), combine instruction-based pretraining with domain adaptation under a structured, curriculum-style regime to further improve specialization. Despite these advances, domain-adaptive pretraining faces fundamental challenges. High-quality curated corpora are often scarce or subject to licensing restrictions , making it difficult to scale adaptation to multiple domains (Wu et al., 2025). Moreover, continual pretraining on narrow domains can cause catastrophic forgetting of general capabilities, as highlighted by recent studies (Béthune et al., 2025; Huang et al., 2024). Thus, while domain-adaptive pretraining improves specialization, it remains constrained by data scarcity and forgetting risks, motivating our work.

Synthetic data generation has recently emerged as a promising alternative with large-scale corpora built from scratch. For instance, Cosmopedia (Ben Allal et al., 2024) uses carefully designed multi-stage prompts to generate diverse open textbook-style pretraining corpora, while the Phi-4 models (Abdin et al., 2024) leverage multi-agent, multi-stage prompting pipelines to produce synthetic datasets spanning hundreds of billions of tokens across diverse domains and skills. These large-scale efforts complement earlier approaches such as Self-Instruct (Wang et al., 2023) and GLAN (Li et al., 2025), which demonstrate that high-quality synthetic instruction–response pairs can improve model generalization, providing scalable alternatives to human-annotated corpora for post-training. Beyond general-purpose synthesis, some methods have lately focused on targeted domain adaptation. For instance, Arannil et al. (2024) mine domain-related subsets from large web datasets, while Yang et al. (2024) generate synthetic text by extracting salient domain entities from documents and constructing connections among them. While such approaches highlight the scalability and diversity benefits of synthetic data, they typically lack systematic coverage of domain concepts or curriculum-aligned progression, which are crucial for instilling principled and progressive domain expertise in LLMs.

The order in which training data is presented has a strong influence on model performance, as first demonstrated by Bengio et al. (2009). Recent work has extended this idea to LLMs. Zhang et al. (2025a) study multiple curriculum strategies for pretraining, including vanilla ordering, pacing-based sampling, and interleaved curricula across six difficulty metrics, showing that thoughtful sequencing can improve training efficiency and downstream accuracy. Complementarily, Lee et al. (2024) introduce curriculum instruction tuning, where sequencing instruction-response pairs by difficulty improves results across diverse benchmarks without additional computational costs. However, such strategies remain largely unexplored in the context of domain knowledge infusion or synthetic textbook-style corpora. In this work, we investigate curriculum scheduling as a principal mechanism to instill progressive knowledge in domain-adaptive pretraining.

## 3 AUTOMATED CURRICULUM-ENHANCED REGIMEN - ACER

ACER is a multi-stage pipeline designed to transform high-level learning goals into a structured synthetic training corpus. As shown in Figure 1, the process begins by capturing domain intent and audience context, expands this information into a detailed outline, and then produces textbook-

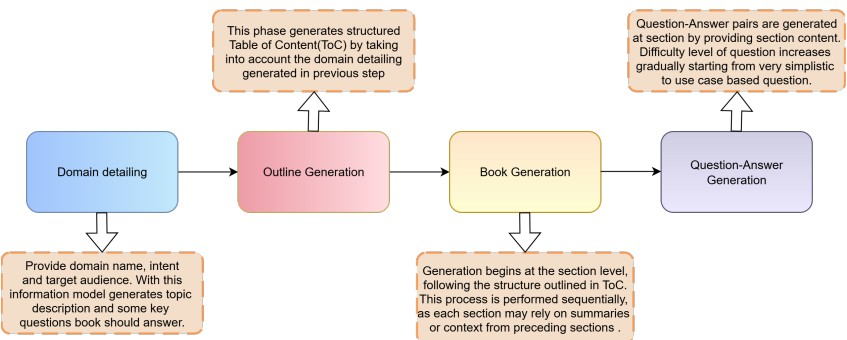

Figure 1: `ACER` - pipeline for synthesis book corpus generation

style sections followed by section-aware question–answer pairs. Each stage feeds the next, ensuring consistency, progressive depth, and factual grounding.

**Domain Detailing**: The first stage of ACER is domain detailing. The goal of this stage is to establish a structured foundation for synthetic corpus generation by capturing domain-specific knowledge requirements, audience context, and intent. This stage aims to formalize the scope, granularity, and emphasis of the target content before automated generation begins, ensuring that all downstream artifacts (textbooks, question–answer pairs, and curricula) are aligned with both educational principles and application needs.

The process begins with three primary inputs:

- **Domain or Topic Name:** A concise label identifying the subject area, e.g., Anatomy, Microeconomics, International Law.

- **Intent:** A statement describing the purpose of the synthetic corpus. For instance, training domain experts, creating introductory learning materials, supporting professional certification preparation, etc.

- **Target Audience Metadata:** Learner profile specifying prior knowledge level (e.g., high school, undergraduate, graduate, or researcher) and contextual constraints (e.g., technical rigor, professional applications).

To translate these inputs into a machine-usable representation, we employ a prompt-driven planning step in which a language model, acting as a "domain author", generates three key artifacts:

- **Domain Description:** A concise, yet comprehensive overview of the topic tailored to the target audience, providing a factual baseline for content generation.

- **Core Subtopics:** A list of areas essential for building expertise, ensuring systematic coverage.

- **Key Questions:** A set of 6–8 relevant questions that the textbook should answer, aligning with Bloom's taxonomy objectives, such as comprehension, application, and synthesis.

The output of this stage is a JSON-encoded schema that captures the domain's description, subtopics, and key questions. This structured representation is editable, enabling subject-matter experts to iteratively refine content priorities before subsequent stages. Additional details about this stage are described in Appendix A.2.1. The prompt used to generate domain detailing are present in Appendix A.6 (Code Block: 1)

**Outline Generation**: Once the domain schema is finalized in the domain detailing stage, the next step is outline generation, where the framework transforms structured topic metadata into a detailed and hierarchical Table of Contents (ToC). This step provides a precise blueprint for the creation of synthetic textbooks, ensuring both thematic coverage and logical progression of ideas. In addition to topic name, intent and target audience (as described in domain detailing), the outline generation process uses the following inputs:

1. **Genre and Style Parameters:** Content preferences, including tone, narrative voice, and language style, to ensure readability and audience alignment.

2. **Domain Schema:** Description, core subtopics, and key questions generated in the Domain Detailing stage.

Collectively, these attributes act as guidance signals for the language model, shaping the structure and content depth of the generated outline. More details about the outline generation phase is described in Appendix A.2.2. The prompt used to generate the ToC can be found in the Appendix A.6 (Code Block: 2)

**Synthetic Content Generation**: Following the generation of the Table of Contents (ToC), the next step involves the generation of synthetic textbook-style content. The ToC is used not only as a hierarchical list, but as a structured blueprint that is systematically traversed and expanded into full-fledged instructional material. This process operates at the *section-level granularity*, ensuring that the generated text is pedagogically cohesive while remaining contextually aligned with the surrounding chapters and sections. A detailed description of the procedure and structure is described in Appendix A.2.3.

Following the synthesis of section-level content, we extend the generation pipeline to incorporate question–answer (QA) pairs derived from the generated text. The rationale for this stage is rooted in pretraining needs: question–answer pairs have been shown to be particularly effective in enhancing reasoning capabilities and comprehension during large-scale model training (Cheng et al., 2024). Unlike continuous narrative text, QA pairs explicitly encode interrogative and expository structures, thereby simulating natural pedagogical exchanges and reinforcing the model's ability to navigate knowledge retrieval, explanation, and application.

The QA generation process unfolds in two sequential stages:

1. **Question Generation:** The data generation model is framed as a subject-matter expert and tasked with generating self-contained, educational questions tied to the section content. For the *first question*, the prompt emphasizes simplicity, typically asking for a factual recall or definition-based understanding. For subsequent questions, the difficulty gradually escalates, following a cognitive progression from comprehension to interpretation, analysis, and application Bloom et al. (1956). This progression ensures that the set of questions reflects an increase in cognitive depth, thus mimicking natural instructional scaffolding. Furthermore, the prompt enforces strict relevance and audience alignment, while prohibiting extraneous labels or formatting.

2. **Answer Generation:** Once the question is obtained, a separate model call generates a detailed, educational, and self-contained answer. The answer is grounded strictly in the section content, ensuring factual alignment and eliminating hallucinations. The model is prompted to produce responses that are rich in detail, employ clear language, and illustrate concepts with examples. The tone is maintained at an educational and explanatory level, suitable for the designated target audience.

The generated question answer pairs are then divided into easy and hard subsets according to their difficulty. The different prompts used for generating section content, question and answer pairs can be found in Appendix A.6 (Code Blocks: 3, 4, 5 respectively).

## 3.1 ACER - CURRICULUM SCHEDULING

We experiment with four curriculum schedules for incorporating synthetic book corpus into continual pretraining (refer Figure 2):

1. **Flat**: All data (books, easy QA, hard QA) from all domains and personas are presented together without any ordering. The data can come in any order, irrespective of their difficulty level.

2. **Cognitive (Cog)**: Data is structured in increasing cognitive difficulty: Books → Easy QA → Hard QA. Inside each of these sections, there is no restriction on the ordering of the domain or persona. Corpora from different domains and personas are mixed randomly.

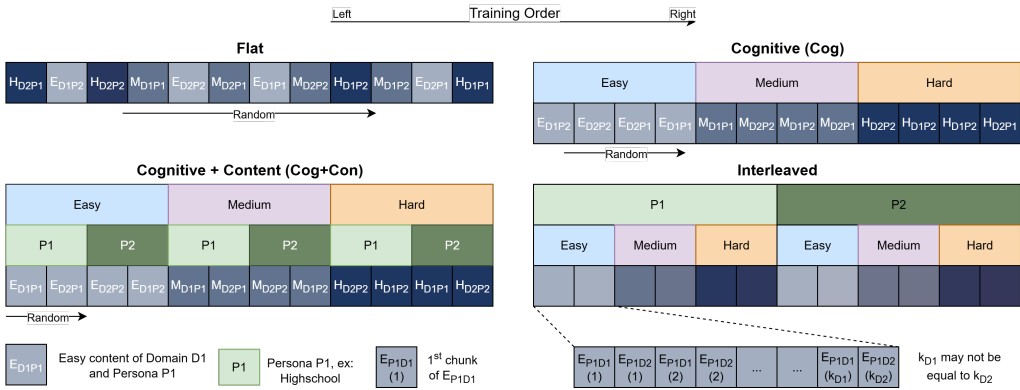

Figure 2: Different training schedules. Personas $P1$ and $P2$ are shown here only for illustration and do not represent the full set of personas. Persona $P2$ lies higher than $P1$ on the cognitive axis (e.g., P2=Undergraduate, P1=high school). $E_{P1D1}(1)$ denotes the first chunk of $E_{P1D1}$, which is formed by combining multiple consecutive sections.

3. **Cognitive + Content (Cog+Con)**: Extends the cognitive schedule by introducing content-based ordering: High school $\rightarrow$ Undergraduate $\rightarrow$ Graduate $\rightarrow$ Researcher. Again, domains can appear in any order and do not need to follow the same sequence across different personas.

4. **Interleaved**: Inspired by Lee et al. (2024), the data is interleaved across domains at the chapter–section level (e.g., Chapter 1, Section 1 from Domain 1 $\rightarrow$ Chapter 1, Section 1 from Domain 2 $\rightarrow \cdots$). This interleaving is applied separately for books, easy QA, and hard QA, while preserving the persona order within each category. The model first encounters high-school content, followed by undergraduate, graduate, and researcher material, with section-level interleaving for each persona. This arrangement prevents the model from seeing all data from a single domain consecutively. Instead, it alternates between sections from multiple domains while respecting both cognitive difficulty and persona order, following a fixed cyclic pattern. This design enables us to test whether continual pretraining with structured sequencing of synthetic knowledge improves model quality.

## 4 EXPERIMENTS

We experiment with Llama 3.2 1B and 3B models, which are compact student models trained using knowledge distillation from larger Llama 3.1 teachers (8B and 70B).

To identify domains where domain specific knowledge infusion is needed the most, we measured the per-domain accuracy gap between the 3B student and its 8B teacher on the full set of 56 MMLU tasks under 0-shot evaluation. Figure 3 shows the difference in accuracy between the student and teacher, with domains ranked by gap size. The largest regressions occurred in microeconomics, statistics, econometrics, mathematics, and psychology. These domains serve as our primary infusion targets. For consistency, we use this same set of domains for both the 1B and 3B students, while treating all other MMLU domains as non-targets. Our evaluation demonstrates that ACER improves model accuracy on these target domains without adversely impacting model's accuracy on non-target domains.

Our synthetic book corpus consists of detailed, domain-specific books that span across diverse topics, each accompanied by exam-style question–answer (QA) pairs. The QA sets are divided into two difficulty levels: easy and hard. For each target domain, we generate synthetic corpora across four audience (persona) levels: high school, undergraduate, graduate, and researcher. To avoid benchmark contamination[1], we decontaminate the corpus by removing any text with cosine similarity greater than 0.9 to MMLU content. Appendix A.1 summarizes the token distribution across

---

[1] Appendix A.7 details the decontamination process showing examples of text fragments with high similarity to MMLU content

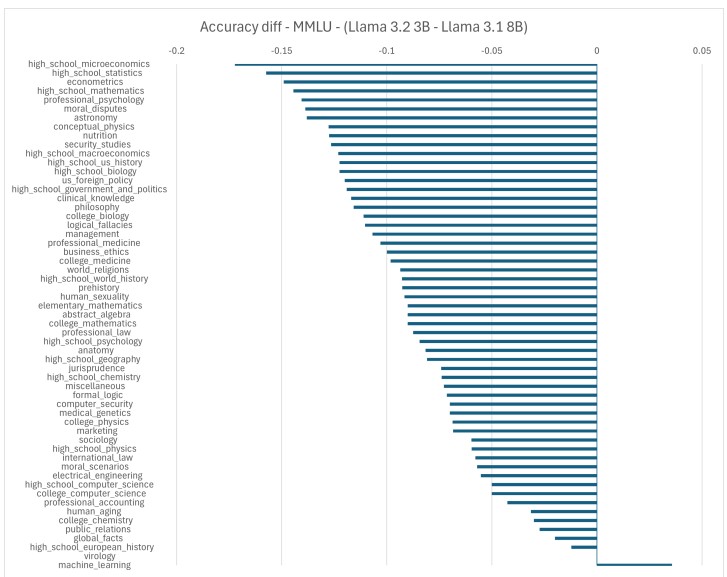

Figure 3: Target domains for knowledge infusion: identified using accuracy difference between the Llama 3.2 3B student model and its teacher in Llama 3.1 8B

domains and audience levels: on average, `ACER` generates 6.4M tokens per domain, totaling about 32M tokens overall. Appendix A.4 lists the generated artifacts for microeconomics domain.

## 4.1 TRAINING DETAILS

We initialize from the pretrained Llama 3.2 1B and 3B checkpoints and continually pretrain them on `ACER` generated synthetic book corpora using the standard next token prediction objective. To mitigate catastrophic forgetting, the synthetic in-domain corpus is mixed 1:1 with general replay data[2]. The key hyper-parameters for training are set as: batch size $= 512$, maximum sequence length $= 8192$, learning rate $= 2 \times 10^{-5}$ with cosine decay to $2 \times 10^{-6}$, and a warm-up of 1% to peak learning rate. The replay data is drawn from pile knowledge, cosmopedia, ultratextbooks, and subsets of proof-pile-2 (openwebmath, algebraic stack, and arXiv).

For evaluation, we focus on five target MMLU subsets: high school microeconomics (MEco$_{hs}$), high school statistics (Stats$_{hs}$), econometrics (Econ), high school mathematics (Maths$_{hs}$), and professional psychology (Psych$_p$). We report both per-domain accuracy and the macro-average across these five subsets, denoted Macro$_t$. To track generalization, we also report macro-average accuracy across the remaining 51 MMLU subsets, denoted as Macro$_{nt}$. Beyond MMLU, we evaluate the models on ARC, GPQA, AGIEval, GSM8K, and HellaSwag to assess broader reasoning and knowledge capabilities.

We ablate across multiple curriculum schedules during continual pretraining, as described in section 3. As a baseline, we use a **Flat** schedule with no ordering to test whether domain-specific content alone improves model performance. We then introduce a **Cognitive (Cog)** schedule that progresses from textbooks → easy QA → hard QA. Next, we add the content dimension, ordering by audience levels (high school → undergraduate → graduate → researcher) in addition to the cognitive dimension; we refer to this as **Cog+Con**. Finally, we evaluate an **Interleaved** schedule that enforces ordering at the chapter-section level. These schedules allow us to test whether structured ordering improves domain knowledge infusion compared to random mixing.

## 4.2 RESULTS: IMPACT OF `ACER` WITH LLAMA 3.2 3B

We generate synthetic book corpora for the identified target domains and continually pretrained the baseline Llama 3.2 3B model using different curriculum schedules. Table 1 compares these trained

---

[2]Appendix A.3 details our ablation study showing 1:1 ratio provides the right tradeoff

Table 1: Comparison of pretrained Llama 3.2 models (3B and 1B) with our knowledge-infused variants trained under different curriculum schedules. Domain-targeted synthetic textbooks with exam-style QA consistently improve the pretrained baselines across all schedules. The **Cog+Con** schedule, achieves the best results, with improvements of 3.0 and 2.6 percentage points in target domains for the 3B and 1B models, respectively. Performance on non-target domains also improves by roughly 0.5 points due to cross-domain knowledge transfer.

| Model | $MEco_{hs}$ | $Stats_{hs}$ | Econ | $Maths_{hs}$ | $Psych_p$ | $Macro_t$ | $Macro_{nt}$ |
|---|---|---|---|---|---|---|---|
| Llama 3.2 3B | 0.5378 | 0.3796 | 0.3158 | 0.2704 | 0.5507 | 0.4108 | 0.5754 |
| Flat | 0.5588 | 0.3843 | 0.3596 | 0.3111 | 0.567 | 0.4362 | 0.5824 |
| Cog | 0.5966 | 0.3796 | 0.3596 | 0.2963 | 0.5686 | 0.4401 | 0.5809 |
| Cog+Con | 0.584 | 0.4028 | 0.3509 | 0.2889 | 0.5768 | **0.4407** | **0.5821** |
| Interleaved | 0.5798 | 0.3611 | 0.3509 | 0.2593 | 0.5605 | 0.4223 | 0.5766 |
| Llama 3.2 1B | 0.3361 | 0.25 | 0.193 | 0.2296 | 0.3448 | 0.2707 | 0.4011 |
| Cog+Con | 0.3445 | 0.3287 | 0.2105 | 0.2481 | 0.3562 | **0.2976** | **0.4051** |

models with the pretrained 3B baseline. Across all schedules, ACER consistently improves performance on the target domains. The **Flat** regime, which uniformly mixes books and QA pairs without ordering, yields a strong gain of +2.5 points in $Macro_t$ over the baseline. Building on this, the **Cognitive (Cog)** schedule, which progresses from books → easy QA → hard QA, provides further incremental improvements. Extending this with persona-based content progression in **Cog+Con** delivers the best overall performance, indicating that curriculum design amplifies the benefits of synthetic corpora in continual pretraining.

In contrast, the **Interleaved** schedule, which mixes book sections and QA pairs across domains within training stages, underperforms relative to the other curriculum schedules, with particularly large drops in mathematics and statistics. This contrasts with Lee et al. (2024), where interleaving educational content improved performance of the model in the fine-tuning phase. We conjecture that, while interleaving may be effective for instruction-following fine-tuning, its fragmented sequencing dilutes the supervision signal in continual pretraining. For smaller-scale domain infusion tasks, such as ours, frequent task-switching introduced by interleaving likely overwhelms model capacity, leading to degraded performance rather than gains.

As shown in Table 1, domain knowledge infusion through targeted continual pretraining provides the largest benefits in microeconomics and econometrics. These niche areas are relatively underrepresented in the pretraining corpus of the 3B baseline, leaving substantial room for improvement[3]. By contrast, domains such as statistics and mathematics are more prevalent in web-scale text, so the baseline already demonstrates strong competence, resulting in only modest gains. Overall, we observe an improvement of approximately 3 percentage points in $Macro_t$. Notably, our training regimen not only prevents catastrophic forgetting but also facilitates positive cross-domain knowledge transfer, improving performance on non-target domains ($Macro_{nt}$) by 0.7 percentage points.

### 4.2.1 RESULTS: SCALING TO SMALLER MODELS- LLAMA 3.2 1B

We also apply the best-performing **Cog+Con** schedule to the Llama 3.2 1B model. Despite its smaller capacity, the 1B student also benefits from knowledge infusion, achieving gains of more than 2.5 percentage points on target domains and 0.4 points on non-target domains compared to its baseline (Table 1). These results highlight that our framework is effective across scales, improving both specialized knowledge and general performance even in compact models.

### 4.3 IMPACT BEYOND MMLU: GENERALIZATION TO BROADER CAPABILITIES

Next, we ask how the proposed knowledge-infusion recipe in ACER transfers beyond MMLU. To this end, we evaluate the knowledge-infused Llama 3.2 models against their pretrained baselines on standard language understanding benchmarks. Table 2 reports results on ARC, GPQA, AGIEval,

---

[3]A.5 shows an example of how ACER helped the model correctly answer a microeconomics question that it had previously answered incorrectly

Table 2: Evaluation across benchmarks comparing our knowledge-infused Llama 3.2 models (3B and 1B) with their pretrained versions. The knowledge-infused models improve tasks requiring knowledge recall and understanding (ARC, GPQA, and MMLU) by more than 2 absolute percentage points, without regressing on general capabilities such as language understanding, reasoning, and mathematics (AGIEval, GSM8K, and HellaSwag).

| Benchmark | #shots | Setting | Llama 3.2 3B | | Llama 3.2 1B | |
|---|---|---|---|---|---|---|
| | | | Pre-trained | Ours | Pre-trained | Ours |
| ARC Challenge | 25 | Acc (Weighted) | 0.4701 | **0.4837** | 0.3652 | **0.3831** |
| GPQA | 0 | Acc (Mean) | 0.2656 | **0.2879** | **0.2545** | 0.2478 |
| MMLU | 5 | Acc (Weighted) | 0.5408 | **0.569** | 0.3099 | **0.3277** |
| AGIEval | 0 | Acc (Weighted) | 0.2255 | 0.2253 | 0.1867 | 0.1877 |
| GSM8K | 5 | Strict (EM) | 0.2805 | 0.2798 | 0.0667 | 0.0591 |
| GSM8K | 5 | Flexible (EM) | 0.2767 | 0.276 | 0.0644 | 0.0523 |
| HellaSwag | 0 | Acc (Mean) | 0.5522 | 0.5489 | 0.4777 | 0.4782 |

GSM8K, and HellaSwag. Tasks that emphasize domain-specific knowledge and recall, such as ARC, GPQA, and MMLU (5-shot), benefit the most. Both the 3B and 1B variants achieve gains of more than two absolute percentage points, highlighting the effectiveness of targeted continual pretraining in strengthening reasoning over underrepresented knowledge areas.

Equally important, these improvements do not come at the cost of general capabilities. On benchmarks such as AGIEval, GSM8K, and HellaSwag, which measure general reasoning, arithmetic, and commonsense capabilities, the ACER models remain stable relative to their pretrained baselines. For the 3B model, differences are within 0.3 absolute points on HellaSwag and even smaller across other tasks, while the 1B model shows comparable stability. This indicates that domain infusion not only enhances specialized competence but also preserves broad language abilities, demonstrating the scalability of our framework across diverse evaluation settings.

## 5 LIMITATIONS OF THE WORK

Although our approach relies on a strong model (Gemini 2.0 Flash) for content generation, we did not ablate between different models for content generation. Additionally, due to time and resource constraints, we evaluated the approach on 1B and 3B scale models without extending the evaluation to larger-scale models. Given that pretraining also relies on the underlying model capacity, more powerful and generalizable models may yield greater benefits.

## 6 REPRODUCIBILITY DETAILS

The parameters required to reproduce our results—such as token budgets, sequence lengths, batch sizes, optimizer settings, learning rate schedule, training steps, hardware specifications, random seeds, prompts, and de-duplication thresholds—are provided in section 4.1. The prompts used for all types of generations are listed in Appendix A.6.

## 7 CONCLUSION

We present ACER (Automated Curriculum-Enhanced Regimen), a novel framework that systematically infuses domain-specific knowledge into large language models through progressive, curriculum-aligned learning sequences interspersed with targeted assessment. Our empirical evaluation demonstrates that ACER achieves substantial performance gains in target domains while maintaining model capabilities across existing tasks and enabling beneficial cross-domain knowledge transfer. This work establishes a principled foundation for structured knowledge integration in LLMs, offering a scalable pathway toward more capable and domain-aware language models.

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

# A APPENDIX

## A.1 TOKENS GENERATED USING ACER SYNTHESIS FLOW

Table 3 details the number of tokens generated by ACER. It uses Gemini 2.0 Flash apis for synthetic data generation.

Table 3: ACER book corpus: #tokens (in Millions) across the five domains and four personas (as generated by Llama tokenizer). Total synthetic data #tokens = 31.97 Millions

| Persona | Microeconomics | Statistics | Econometrics | Mathematics | Psychology |
|---|---|---|---|---|---|
| Highschool | 0.67 | 0.74 | 0.55 | 1.04 | 0.72 |
| Undergraduate | 1.56 | 1.47 | 1.67 | 1.97 | 1.01 |
| Graduate | 2.56 | 2.10 | 2.73 | 1.55 | 1.27 |
| Researcher | 1.66 | 1.79 | 2.83 | 2.21 | 1.85 |
| Total | 6.46 | 6.10 | 7.78 | 6.77 | 4.85 |

## A.2 ACER SYNTHESIS FLOW

### A.2.1 DOMAIN DETAILING

Explicitly encoding domain detailing before corpus synthesis offers two advantages:

1. Provides a transparent knowledge map that facilitates traceability between input requirements and generated content.
2. Enables curriculum-aware scaling, where subsequent synthetic data creation (e.g., textbooks and assessments) follows a structured hierarchy rather than ad-hoc sampling.

### A.2.2 HIERARCHICAL STRUCTURE GENERATION

ACER leverages a large language model prompted as a professional author to create a **multilevel outline** in a structured JSON format. The generated outline includes the following.

- **Book Title:** A descriptive title aligned with the topic and purpose.
- **Parts:** Four to six major parts that divide the book into thematic areas, each with a concise summary.
- **Chapters:** Four to six chapters per part, each providing a self-contained exploration of subtopics.
- **Sections and Subsections:** Three to six sections per chapter, with optional subsections for complex concepts, ensuring fine-grained coverage.

This structured representation balances clarity, depth, and scalability, allowing the same framework to create outlines of varying complexity based on audience needs.

Explicit outline generation provides three main benefits:

1. **Consistency:** Maintains structural uniformity across different domains and audience levels.
2. **Scalability:** Simplifies automation, as the JSON schema can be directly consumed by subsequent content generation pipelines.
3. **Pedagogical Alignment:** Encourages systematic progression from foundational concepts to advanced material, supporting curriculum-driven pretraining.

### A.2.3 SYNTHETIC CONTENT CREATION

The procedure begins with the construction of a tree representation from the ToC. Each node in this tree corresponds to a textual unit—either a *Part*, *Chapter*, *Section*, or *Subsection*. For every node,

the system maintains key attributes that include the title, description, content, and node type (an enumerated label designating whether the node belongs to root, part, chapter, section, or subsection). This representation provides a flexible yet structured foundation for downstream content synthesis.

Once the hierarchical structure is established, the system proceeds to parse the tree and generate section-level content. For each node in the section, the input prompt is carefully composed to include contextual information such as the title and description of the enclosing part, the title and description of the chapter, the focal section title, and the list of subsections under it. In addition, metadata style guidelines (intent, audience, genre, tone, voice, and language) are explicitly injected into the prompt. This ensures that the generation process is not only content-driven but also stylistically aligned with the pedagogical and editorial goals of the textbook. The LLM is instructed to produce polished, instructional prose while avoiding any meta-commentary, annotations, or explanations of its reasoning.

### A.3    ABLATIONS: WHAT IS THE RIGHT DATA MIXTURE?

Table 4 reports the effect of varying the ratio of in-domain synthetic data to replay data on model performance under the Cog+Con curriculum. The balanced 1:1 setting yields the strongest results, improving $Macro_t$ by about 3 absolute points over the baseline and $Macro_{nt}$ by 0.7 points. Ratios skewed toward replay (1:3) or synthetic data (3:1, 9:1) yield smaller gains and fail to surpass the balanced mixture. The trend is particularly evident for non-target domains: as replay proportion decreases, $Macro_{nt}$ consistently drops, underscoring the role of replay in preserving generalization. Conversely, in-domain synthetic data is the main driver of improvements on target domains. Overall, the balanced 1:1 mixture offers the best trade-off between specialization and generalization, pointing to adaptive mixture schedules as a promising future direction.

Table 4: Ablation on in-domain to replay data proportion for training: CPT with equal proportion of in-domain and replay data provides the right balance

| Ratio | Baseline | 1:3 | 1:1 | 3:1 | 9:1 |
|---|---|---|---|---|---|
| Target Macro | 0.4108 | 0.4257 | **0.4407** | 0.4284 | 0.4135 |
| Non-Target Macro | 0.5754 | 0.5759 | **0.5821** | 0.5805 | 0.5736 |

### A.4    EXAMPLE DOMAIN: MICROECONOMICS

In this section, we look at an example from **Microeconomics** Domain, we see snippets of the following:

- Topic detailing
- ToC (shortened version)
- Example section
- Example QA Pair
- Example Easy QA Pair
- Example Hard QA Pair
- A snapshot of the ToC Tree Visualiser

**Topic Detailing:** Imagine you're running a lemonade stand. Microeconomics is like understanding all the tiny decisions that make your stand successful 2013 how many lemons to buy, what price to charge for your lemonade, and whether to hire a friend to help you out. It's about how individuals, families, and businesses make choices about using limited resources. Instead of looking at the entire economy of a country (that's macroeconomics!), microeconomics zooms in on these smaller, everyday economic activities. It helps us understand why things cost what they do, how markets work, and how people respond to different incentives. Think of it as the science of small-scale economic decisions and their ripple effects.

**Shortened ToC**

```
{
  "title": "Part 1: The Basics of Making Choices",
  "description": "This section introduces the foundational.... ",
  "chapters": [
    {
      "title": "Chapter 1: Welcome to Microeconomics!",
      "description": "This chapter introduces the world of microeconomics ...",
      "sections": [
        {
          "title": "What is Microeconomics?",
          "subsections": [
            {
              "title": "The 'Micro' in Microeconomics"
            },
            {
              "title": "Lemonade Stands and Economies"
            }
          ]
        },
        {
          "title": "Why Should You Care About Microeconomics?",
          "subsections": [
            {
              "title": "Making Smarter Choices"
            },
            {
              "title": "Understanding the World Around You"
            }
          ]
        },
        .
        .
      ]
    }
  ]
}
```

**Example Section:** Scarcity, as we've established, means we can't have everything we want. This universal truth forces us to confront some tough questions. If we can't satisfy all our desires, how do we decide which ones to satisfy and which ones to leave unfulfilled? The answer lies in two closely related concepts: prioritization and rationing. ...

**Example QA**:
**Question**: What are the three main decision-makers studied in microeconomics?
**Answer**: Microeconomics focuses on the "small picture" of the economy, studying the decisions made by individual players rather than the entire national or global economy.....

**Example Easy QA**:
**Question**: Imagine you're deciding whether to spend your allowance on a new video game or save it for a concert ticket. How does this single decision illustrate the core idea that microeconomics studies choices made by individuals with limited resources?
**Answer**: Okay, that's a great question! It gets right to the heart of what microeconomics is all about. Let's break it down using the lemonade stand example and the ideas presented in the text......

**Example Hard QA**:
**Question**: Imagine two lemonade stands on the same street. One stand uses only organic lemons and charges a higher price, while the other uses regular lemons and charges a lower price......
**Answer**: Okay, that's a great question! It gets right to the heart of how microeconomics works in the real world, even in something as simple as two lemonade stands. Microeconomics helps us

understand why both lemonade stands 2013 the one with organic lemons and a higher price and the.....

**A snapshot of the Tree Visualiser Tool**, which used for analysing the Generated Data.

Figure 4: Snapshot of the Tree Visualisation tool, we created to analyse the ToC and the associated generated Data

## A.5 EXAMPLE OF PERFORMANCE IMPROVEMENT AFTER ACER

We present an example MMLU question from the high-school microeconomics subset to demonstrate the effectiveness of ACERThe base pre-trained model (LLaMA3.2-3B) could not answer this question correctly, while the ACER trained model was able to find the right answer due to its enhanced domain understanding.

**Question:** "The elasticity of supply is typically greater when ..."

**Options:**

    A. Producers have fewer alternative goods to produce.

    B. Producers have less time to respond to price changes.

    C. Producers are operating near the limits of their production.

    D. Producers have more time to respond to price changes.

The correct answer is option D. The base LLaMA3.2-3B model predicted option A (incorrect), while our CPT-enhanced model predicted option D (correct).

This improvement can be attributed to the additional knowledge injected during continual pretraining. Our generated microeconomics book included a dedicated section on *"Supply Curve and Elasticity"* An excerpt from the Table of Contents of that book is shown below:

```
{
  "title": "Advanced Producer Theory: Cost and Production",
  "description": "This chapter focuses on the theory of the firm...",
  "sections": [
    {
      "title": "Profit Maximization and Supply",
      "subsections": [
        {
          "title": "Supply Curve and Elasticity"
```

```
                    }
                ]
            },
            {
                "title": "Applications of Producer Theory",
                "subsections": [
                    {
                        "title": "Impact of Technological Change"
                    }
                ]
            },
            {
                "title": "Summary"
            }
        ]
    }
```

## A.6 PROMPTS USED IN THE KNOWLEDGE INFUSION FRAMEWORK

### A.6.1 TOPIC DETAILING PROMPT

The prompt used for topic detailing, which will be used for ToC generation.

```
1  prompt = f"""
2      You are an expert author preparing to write a comprehensive book.
3      Topic:\"{topic}\".
4      Target audience: \"{audience}\".
5      Intent (purpose): \"{intent}\".
6
7      Your task is to generate preparatory material that will help you
          structure the book. Based on the above, provide the following:
8      1. A factual and high-level description of the topic, suitable for
          the target audience.
9      2. A list of 6 8   core themes or subtopics that must be covered to
          provide a well-rounded understanding of the topic.
10     3. A list of 6 8   important questions that the book should aim to
          answer from the perspective of the target audience.
11
12     Present your output in JSON with the following keys:
13     - "description": A string
14     - "subtopics": A list of strings
15     - "key_questions": A list of strings
16     """
```

Code Block 1: Prompt construction for Topic Detailing

### A.6.2 TOC GENERATION PROMPT

The prompt used for ToC generation:

```
1
2  rompt = f'''
3  You are a professional book author known for clear, structured, and
      reader-friendly writing.
4  Your task is to design a full book outline based on the information
      below.
5  ---
6  **Topic:** {topic}
7
8  **Target Audience:** {audience}
9
10 **Intent:** {intent}
11
```

```
12  **Genre:** {genre}
13
14  **Tone:** {tone}
15  **Voice:** {voice}
16  **Language Style:** {language}
17
18  **Topic Description:**
19  {description}
20
21  **Core Subtopics to Cover:**
22  {subtopics}
23
24  **Key Questions the Book Should Answer:**
25  {key_questions}
26  ---
27
28  Based on this, generate a comprehensive book structure in JSON with the
       following:
29  1. A compelling and descriptive "title" for the book.
30  2. Divide the book into **4 to 6 major parts**:
31     - Each part should have:
32       - "title" (clear, theme-based)
33       - "description" (3 5  sentence summary)
34  3. Each part should contain **4 to 6 chapters**:
35     - Each chapter should include:
36       - "title"
37       - "description" (3 5  sentences, aligned with purpose)
38       - "sections": 3 to 6 entries, each with:
39          - "title" (section title)
40          - "subsections": Optional (to describe complex ideas, 2-3
                subsections per section)
41             - "title" (subsection title)
42     (Include common sections like "Introduction" and "Summary" where
           appropriate.)
43
44  Return **only** a well-formatted JSON object in the following structure:
45  {{
46    "title": "<book_title>",
47    "parts": [
48       {{
49         "title": "<part_title>",
50         "description": "<paragraph>",
51         "chapters": [
52           {{
53             "title": "<chapter_title>",
54             "description": "<paragraph>",
55             "sections": [
56                {{
57                  "title": "<section_name>",
58                  // Optional subsections
59                  "subsections": [
60                    {{
61                      "title": "<subsection_name>"
62                    }},
63                    ...
64                  ]
65                }},
66                ...
67             ]
68           }},
69           ...
70         ]
71       }},
72       ...
73     ]
```

```
74    }}
75    '''
```

Code Block 2: Prompt construction for ToC generation

### A.6.3 SECTION GENERATION PROMPT

The prompt used for Section Content generation:

```
1
2  prompt= "Generate detailed, high-quality, and cohesive book content for
       the following section. Expand fully on each idea and subsection with
       in-depth explanations, compelling examples, and smooth transitions.
       Follow the style guidelines strictly. Avoid meta-commentary or
       structural notes. Write as if this is the final, polished content
       ready for publication."
3
4  metadata_str = '''Style Guidelines:
5  - Intent: {intent}
6  - Audience: {audience}
7  - Genre: {genre}
8  - Tone: {tone}
9  - Voice: {voice}
10 - Language: {language}
11 - Description: {description}
12 '''
13
14 end_prompt = f'''Instructions
15 - Compose a flowing narrative or exposition that naturally integrates
       the listed subsections.
16 - Be generous with elaboration, examples, and insights.
17 - Ensure thematic and stylistic continuity with the previous content.
18 - Do not include headers, instructions, or artificial markers    only
       write the finished prose.
19 '''
```

Code Block 3: Prompt construction for section content generation

### A.6.4 QA PAIR GENERATION PROMPTS

The prompt used for question generation is:

```
1  prompt = f"""
2  You are a professor specializing in the subject of {topic_name}. Your
       task is to generate **educational, self-contained questions** to
       help students understand a section from a textbook titled:
       "{book_title}".
3  This book is for {target_audience}, with the primary goal to:
       **{intent}**
4
5  ### Global Context Start ###
6  - **Topic**: {topic_name}
7  - **Topic Description**: {topic_description}
8  - **Questions the book attempts to answer**: {guiding_questions}
9  ### Global Context End ###
10
11 You will be generating questions for the following part of the book:
12 - Chapter: {chapter_title}
13 - Section: {section_title}
14 --- Section Content Start ---
15 {section_text}
16 --- Section Content End ---
17 """
18
```

```python
if previous_question:
    prompt += f"""
### TASK ###

You are generating the next question in a sequence.

1. Analyze the difficulty of the previous question.
2. Generate a new question that is **slightly more challenging** than
    the one before. Increase cognitive d e p t h move progressively from
    understanding    interpretation    analysis    application.
3. Ensure the question is different in focus or angle from the previous
    one.
4. Do **not** include the answer.
5. **Strictly** maintain relevance to the given section and suitability
    for {target_audience}.
6. **Only output the question** do  not include any prefixes like
    "Generated Question", "Q1:", or anything else.

Previous Question:
- {previous_question}

### Question ###
Question:"""
else:
    prompt += f"""
### TASK ###

You are generating the **first question** in a learning sequence.

1. Start with a **simple question** focused on factual recall or basic
    definitions.
2. Make the question self-contained and directly based on the section
    content.
3. Do **not** include the answer.
4. Ensure clarity, relevance, and suitability for {target_audience}.
5. **Only output the question** do  not include any prefixes like
    "Generated Question", "Q1:", or anything else.

### Question ###
Question:"""
```

Code Block 4: Prompt construction for question generation

The prompt used for answer generation is:

```python
prompt = f"""
You are a professor specializing in the subject of {topic_name}. Your
    task is to generate an **educational, self-contained answer** to a
    question based on a section from a textbook titled: "{book_title}".
This book is for {target_audience}, with the primary goal to:
    **{intent}**

### Global Context Start ###
- **Topic**: {topic_name}
- **Topic Description**: {topic_description}
### Global Context End ###

You are answering a question based on the following section:
- Chapter: {chapter_title}
- Section: {section_title}

--- Section Content Start ---
{section_text}
--- Section Content End ---
```

```
18  ### TASK ###
19
20  Your job is to write a complete, thoughtful, and educational answer to
        the student's question below. Follow these guidelines:
21
22  1. Ensure the answer is **directly grounded in the section content**
        provided above.
23  2. Make the answer **rich in detail**, using clear language and examples
        when helpful.
24  3. The answer should be **self-contained**, meaning the reader should
        not need to refer to the original content to understand it.
25  4. Ensure the tone is educational and appropriate for {target_audience}.
26  5. Do **not invent any information** not present in the section.
27
28  ### Question ###
29  {question}
30
31  ### Answer ###
32  """
```

Code Block 5: Prompt construction for answer generation

## A.7 DECONTAMINATION- DATA ANALYSIS

We did a semantic deduplication over the synthetic data generated and the benchmark used. The motivation for semantic deduplication is that it is much more thorough over Minhash Based Deduplication. We calculated the embeddings for both generated and benchmark data. We find the closest neighbor for each generated data sample, using cosine similarity metric. We analyse the samples which are above a certain threshold. The behaviour we noticed across domains is that there is little to no contamination with respect to benchmark data. And for the samples whose similarity is above 0.9, the generated samples discuss a set of topics that any one would study while gaining knowledge about the domain (refer Table 5 for decontamination report and Table 6 for example fragments).

Table 5: Domain Statistics with Generated samples (Threshold for Similarity=0.8). The Final column contains the percentage of Benchmark Samples which have similairty >=0.8

| Domain | Generated Samples | % Similarity $\geq$ 0.8 |
|--------|-------------------|-------------------------|
| Econometrics | 4125 | 0.0969 |
| Mathematics | 4110 | 0.0243 |
| Psychology | 4350 | 0.2758 |
| Statistics | 4200 | 0.1428 |
| Microeconomics | 3900 | 0.051282 |

Table 6: Examples of Generated Samples, Closest Benchmark Samples, and Cosine Similarities across Domains

| Contents |
|---|
| **Domain**: Econometrics |
| **Generated Sample:** What is the primary consequence of omitting a relevant variable from a multiple linear regression model? |
| **Closest Benchmark Sample:** If a relevant variable is omitted from a regression equation, the consequences would be that: |
| i) The standard errors would be biased ii) If the excluded variable is uncorrelated with all of the included variables, all of the slope coefficients will be inconsistent. iii) If the excluded variable is uncorrelated with all of the included variables, the intercept coefficient will be inconsistent. iv) If the excluded variable is uncorrelated with all of the included variables, all of the slope and intercept coefficients will be consistent and unbiased but inefficient. |
| **Cosine Similarity:** 0.8456 |
| **Domain**: Mathematics |
| **Generated Sample:** A university club with 15 members needs to form a committee of 5 to organize a fundraising event. * How many different committees can be formed? * The club decides that the committee must have a president, a vice-president, a treasurer, a secretary, and a public relations officer. The president and vice-president must be chosen from among 8 senior members, while the treasurer, secretary, and public relations officer can be chosen from the remaining members (seniors or otherwise). In how many ways can such a committee be formed, ensuring each member has a distinct role? |
| **Closest Benchmark Sample:** How many different possible committees of 5 people can be chosen from a group of 15 people? |
| **Cosine Similarity:** 0.8338 |
| **Domain**: Psychology |
| **Generated Sample:** What is the primary difference between classical conditioning and operant conditioning? |
| **Closest Benchmark Sample:** What is the major difference between classical and operant conditioning? |
| **Cosine Similarity:** 0.9871 |
| **Domain**: Statistics |
| **Generated Sample:** What is a sampling distribution, and how is it constructed? |
| **Closest Benchmark Sample:** What is a sampling distribution? |
| **Cosine Similarity:** 0.929 |
| **Domain**: Microeconomics |
| **Generated Sample:** What fundamental economic problem does microeconomics primarily address? |
| **Closest Benchmark Sample:** The primary focus of microeconomics is |
| **Cosine Similarity:** 0.8312 |

