# OpenReview forum: "From Amateur to Master: Infusing Knowledge into LLMs via Automated Curriculum Learning"
_ICLR.cc/2026/Conference — ICLR 2026 Conference Withdrawn Submission_

### Official Review · Reviewer_WdyZ · 2025-10-24

**Soundness:** 2
**Presentation:** 1
**Contribution:** 2
**Rating:** 2
**Confidence:** 4

**Summary:**

The paper proposes ACER (Automated Curriculum-Enhanced Regimen), a framework for infusing domain-specific knowledge into large language models (LLMs) through automated generation of textbook-style curricula and Bloom’s taxonomy–guided question–answer (QA) pairs. The synthetic data is used for continual pretraining under various curriculum scheduling strategies, including cognitive progression (from textbooks to easy/hard QA) and persona-based content ordering (high school → researcher). Experiments on Llama 3.2 (1B and 3B) show consistent gains (~3 percentage points macro-average) on five MMLU subdomains where the base models underperform relative to larger teachers (e.g., microeconomics, econometrics). The authors also report modest improvements on non-target MMLU domains (+0.7 points), as well as gains on knowledge-intensive benchmarks like ARC (Clark et al., 2018) and GPQA (Rein et al., 2023), without degradation on general reasoning tasks (e.g., GSM8K, HellaSwag).

**Strengths:**

1. Well-motivated problem: The paper addresses a genuine limitation of current LLMs—their shallow understanding of specialized domains—highlighted by consistent performance gaps on benchmarks like MMLU (Hendrycks et al., 2021).
2. Systematic synthesis pipeline: ACER’s multi-stage generation process (domain detailing → outline → textbook → QA pairs) is pedagogically grounded and scalable, drawing on established educational frameworks like Bloom’s taxonomy (Bloom et al., 1956).
3. Comprehensive evaluation: The authors evaluate across multiple benchmarks (MMLU, ARC, GPQA, AGIEval, GSM8K, HellaSwag) and include ablations over curriculum scheduling strategies.
4. Reproducibility: Training details, data mixing ratios, decontamination procedures, and prompt templates are thoroughly documented in the appendix.

**Weaknesses:**

1. Lack of comparison to strong synthetic data baselines: The paper does not compare ACER against recent, high-impact synthetic data methods, like Phi-4 (Abdin et al., 2024), which uses multi-agent, multi-stage pipelines to generate diverse, high-quality textbook-like data. Without such comparisons, it is unclear whether gains stem from curriculum structure or simply from high-quality synthetic data.
2. Marginal and unstable gains from curriculum scheduling: While the “Flat” baseline (random mixing of books and QA) already yields +2.5 points, the best curriculum (Cog+Con) adds only ~0.5 additional points. More concerning, the Interleaved schedule—inspired by Lee et al. (2024)—performs worse than Flat, particularly in mathematics and statistics. This undermines the central claim that structured sequencing is beneficial, and suggests the gains may be schedule-sensitive rather than robust.
3. Heavy reliance on a powerful external model for data generation: The synthetic corpus is generated using Gemini 2.0 Flash, a proprietary, state-of-the-art model. The paper provides no ablation using weaker or open-source generators (e.g., Llama 3 itself). This raises concerns about generalizability: ACER may essentially be a form of knowledge distillation from a stronger teacher, not a self-contained curriculum learning method.
4. Limited model scale and cherry-picked domains: Evaluation is restricted to 1B/3B models. Larger models (e.g., 7B/8B) may already encode sufficient domain knowledge, rendering ACER’s marginal gains irrelevant at scale. Moreover, the five target domains are selected based on the largest student–teacher gaps in MMLU—a reasonable heuristic, but one that risks selection bias; the method’s efficacy on domains with smaller gaps remains untested.
5. Overstated claims about catastrophic forgetting: While non-target MMLU performance improves slightly (+0.7), the 1B model shows degradation on GSM8K (Strict EM: 0.0667 → 0.0591). Although small, this suggests potential capacity interference. The paper lacks more rigorous forgetting metrics (e.g., loss on original pretraining data).

**Questions:**

see weakness

---

> ### Author Response · Authors · 2025-11-24
> **Review response**
>
> We thank the reviewer for his/her invaluable time and constructive feedback. Below we address the weaknesses and/or questions point-by-point.
>
> 1. Lack of comparison to strong synthetic data baselines
> 	- We agree that including strong synthetic-data baselines will strengthen the paper. We will add results using a Phi-4 baseline. In the current version, we relied solely on Gemini 2.0 Flash for data generation due to cost and time constraints, which is why other generators were not included. In the revision, we will also ablate with open-source generators to provide a clearer picture of whether the gains arise primarily from the high-quality synthetic data or from the curriculum structure itself.
>
> 2. Marginal and unstable gains from curriculum schedules
> 	- The Flat schedule already incorporates high-quality textbooks and QA pairs, explaining the strong +2.5 percentage point baseline. On top of this, curriculum gains (+0.4 to +0.6) are smaller but consistent.
> 	- Interleaved underperforms because it fragments conceptual continuity. Unlike Lee et al. (2024) which interleaves instruction-tuning tasks, our CPT setting uses long-horizon domain texts. We will add an analysis explaining why frequent domain switching hurts CPT but not instruction tuning.
>
> 3. Heavy reliance on a powerful external model for data generation:
> 	- We acknowledge this limitation. Due to compute constraints, ablations with weaker generators were not included initially. We will add an ablation using an open-source generator (Llama 3.1 8B) to quantify dependence on teacher strength.
>
> 4. Limited model scale and cherry-picked domains:
> 	- We selected target domains using a principled criterion: the largest student–teacher gaps across all 56 MMLU domains. To reduce selection bias, after identifying top-3 gaps, we included two additional domains to broaden coverage (total 5 domains). For domains with small gaps, diminishing returns are expected. We will clarify this reasoning and include a short discussion on transferability to domains with smaller gaps.
>
> 5. Overstated claims about catastrophic forgetting:
> 	- To show generalization beyond MMLU, we evaluated the ACER-trained models on a range of other tasks. Our claims are that ACER improves performance (by about 2 percentage points) on tasks requiring knowledge recall and understanding, and that for tasks focused on language understanding, reasoning, and math, the models remain stable. As the reviewer notes, the ACER-trained 1B model shows a slight degradation relative to the pretrained baseline. However, taken collectively, the differences between the pretrained and ACER-trained models are not significant. Our claims were intentionally conservative, and we will revise the corresponding language to avoid any overstatement and explicitly acknowledge the potential for interference at smaller model scales.

---

> > ### Comment · Reviewer_WdyZ · 2025-11-26
> >
> > I appreciate the authors' responses. However, given that the authors could not provide a revised manuscript that contains their improved results, I decided to keep my score.
> > Here are some suggestions that the authors could use to further improve their paper:
> > 1. Font size in Figure 1 is too small.
> > 2. Figure 2 is not self-contained and is hard to understand.
> > 3. Font size in Figure 3 is too small. Moreover, the major experiment part could not be limited to just one dataset.

---

### Official Review · Reviewer_911Q · 2025-10-30

**Soundness:** 2
**Presentation:** 2
**Contribution:** 1
**Rating:** 2
**Confidence:** 3

**Summary:**

The authors introduce ACER, a framework that synthesizes textbook-style corpora alongside complementary exam-style question-answer pairs. They also design curriculum learning methods that align both cognitive and content dimensions.

**Strengths:**

The paper identifies that state-of-the-art LLMs struggle in specialized domains requiring deep, principled understanding.

**Weaknesses:**

- The paper only compares the effectiveness of different curriculum schedules within their own method in Table 1, and lacks comparisons against other baseline approaches in the same domain (e.g., those referenced in lines 53–55).

- The ACER synthesis seems to rely on Gemini 2.0 Flash as the LLM teacher, but this is not explicitly stated in Section 3. Moreover, the paper omits presenting Gemini 2.0 Flash’s performance on the evaluation benchmarks, which is important as it is the teacher model.

- The ACER simulations lack input from external domain knowledge, as the authors’ emphasis on the need for “principled domain expertise.”

- The paper does not include results from larger open-source or proprietary models, even as reference results.

**Questions:**

- Please add results of strong data-curriculum baselines (e.g., self-instruct,and works cited in lines 53–55).

- State explicitly which teacher LLM is used for ACER synthesis (it seems to be Gemini 2.0 Flash).

- Report the teacher's and other LLMs' scores on all benchmarks

- Describe quality controls for factual/conceptual soundness of the synthesized corpus.

---

> ### Author Response · Authors · 2025-11-24
> **Review response**
>
> We thank the reviewer for his/her invaluable time and constructive feedback. Below we address the weaknesses and/or questions point-by-point.
>
> 1. The paper only compares the effectiveness of different curriculum schedules within their own method in Table 1, and lacks comparisons against other baseline approaches in the same domain (e.g., those referenced in lines 53–55).
> 	- Self-Instruct, GLAN, and related pipelines (e.g., those referenced in lines 53–55) are optimized for instruction tuning rather than structured textbook synthesis. To the best of our knowledge, we have not come across any prior work that performs domain-specific synthetic book creation. Nevertheless, we agree that baseline comparisons are important. For completeness, we will compare our method with the Phi-series of models, which use large-scale synthetic textbook-style data, and include these results in the revised version.
>
> 2. The ACER synthesis seems to rely on Gemini 2.0 Flash as the LLM teacher, but this is not explicitly stated in Section 3. Moreover, the paper omits presenting Gemini 2.0 Flash’s performance on the evaluation benchmarks, which is important as it is the teacher model.
> 	- We agree. Section 3 will be updated to explicitly state Gemini 2.0 Flash as the generator. We will also report its performance on all evaluation benchmarks for transparency.
>
> 3. The ACER simulations lack input from external domain knowledge, as the authors’ emphasis on the need for “principled domain expertise.”
> 	- Our approach deliberately explores the minimal-supervision regime where only generative structure (not external knowledge graphs or curated corpora) guides synthesis. We will update the text to clarify this design choice and add a discussion on how external knowledge could be integrated in future work.
>
> 4. Missing larger-model reference results
> 	- We will include results for Llama 3.1 8B (student-teacher baseline) as a reference point to contextualize the 1B/3B gains.

---

### Official Review · Reviewer_sBBz · 2025-11-02

**Soundness:** 2
**Presentation:** 3
**Contribution:** 1
**Rating:** 2
**Confidence:** 4

**Summary:**

This paper describes a new method for improving domain-expertise in LLMs. Unlike prior work where synthetic instruction data or domain-specific pretraining lacked structured progression, the authors propose ACER (Automated Curriculum-Enhanced Regimen), which automatically generates textbook-style content plus question-answer pairs following Bloom’s taxonomy, and uses a curriculum-aware schedule (cognitive difficulty + persona audience progression) to continually pretrain a general LLM. The proposed method is evaluated on subsets of the MMLU benchmark (five niche domains), as well as ARC, GPQA and other tasks. Experiment results show 3 percentage point macro-average improvement in target domains while preserving general capabilities and achieving 0.7 point gains in non-target domains.

**Strengths:**

1. The paper presents a systematic synthetic corpus generation pipeline that is well-motivated and clearly described.
2. The curriculum scheduling is a reasonable design choice, enabling ablations that highlight the value of ordering in continual pretraining.
3. Empirical results demonstrate improvements in both target niche domains and stability on general capability benchmarks.

**Weaknesses:**

1. The impact of the synthesis book corpus generation pipeline is not sufficiently discussed. The experiments centered around using the same pipeline under different curriculum schedules. It's unclear how big a role the generation pipleline plays in the overall performance improvement.
2. Limited insight was revealed and discussed among the different curriculum schedules, e.g. what makes some schedule works better than the others.
3. The performance improvement in some domains are rather limited and may well fell within the range of variance, e.g. Econ, psych, and Macro_nt.

**Questions:**

1. What's the impact of the pipeline design for the synthesis book corpus generation?
2. Any insight on how sensitive are the results to the quality of the synthetic textbook content, e.g. if generated from a weaker model?

---

> ### Author Response · Authors · 2025-11-24
> **Review response**
>
> We thank the reviewer for his/her invaluable time and constructive feedback. Below we address the weaknesses and/or questions point-by-point.
> 1. The impact of the synthesis book corpus generation pipeline is not sufficiently discussed.
> 	- Our synthetic book generation pipeline is indeed the primary source of new information the model receives. All curriculum schedules operate on identical corpora; therefore, the performance gains observed across all schedules, including the Flat baseline (+2.5 points), demonstrate that the pipeline itself contributes substantially to the improvement. The curriculum schedules serve to further optimize how this corpus is consumed. We will clarify this distinction and provide additional discussion on the pipeline’s role.
>
> 2. Limited insight was revealed and discussed among the different curriculum schedules, e.g. what makes some schedule works better than the others.
> 	- We will expand Section 4.2 to include qualitative insights. In short, schedules that preserve coherent conceptual progression (Cog, Cog+Con) yield more stable improvements, while those that fragment supervision signals (Interleaved) appear detrimental at CPT scale. We will include a deeper analysis of these interactions.
>
> 3. The performance improvement in some domains are rather limited and may well fell within the range of variance, e.g. Econ, psych, and Macro_nt.
> 	- The absolute gains observed are: Psychology +2.61%, Economics +3.51%, and Macro (non-target) +0.67%.
> 	- For the target domains (Psychology, Econ), these exceed typical 3B-scale MMLU variance. Macro_nt is an average across non-target domain; improvements there arise from cross-domain transfer. We will clarify variance estimates and cross-domain transfer discussion.
>
> 4. Any insight on how sensitive are the results to the quality of the synthetic textbook content, e.g. if generated from a weaker model?
> 	- Due to cost constraints, we used Gemini 2.0 Flash exclusively. We will explicitly acknowledge this limitation and add a short experiment comparing a small open-source generator (e.g., Llama 3.1 8B) to evaluate sensitivity.

---

### Note · Authors · 2026-01-03

**Comment:**

We are improving our work with additional experiments and planning to submit the revised manuscript to next available NLP/ML conference.

**Withdrawal Confirmation:**

I have read and agree with the venue's withdrawal policy on behalf of myself and my co-authors.